



# Comparison of atmospheric CO, CO$_2$ and CH$_4$ measurements at Schneefernerhaus and the mountain ridge at Zugspitze

Antje Hoheisel[1], Cedric Couret[2], Bryan Hellack[2], and Martina Schmidt[1]

[1]Institute of Environmental Physics, Heidelberg University, Heidelberg, Germany
[2]German Environment Agency UBA, Germany

**Correspondence:** Antje Hoheisel (antje.hoheisel@iup.uni-heidelberg.de)

**Abstract.** The CO, CO$_2$, and CH$_4$ mole fractions have been measured since 2002 at the Environmental Research Station Schneefernerhaus (ZSF), which is located approximately 300 m below the summit of Mount Zugspitze. Although the station is located remotely at an altitude of 2666 m a.s.l., local pollution events by snow blowers and snow groomers can be detected in the high temporal resolution time series of seconds or minutes. Therefore, a time-consuming flagging process, carried out

manually by the station manager, is necessary.

To examine local influences and the effectiveness of data flagging, a 290 m long intake line to the higher Zugspitz ridge was used to measure CO, CO$_2$ and CH$_4$ mole fractions at a potentially less polluted location between October 2018 and October 2020. The comparison of these two time series shows that the mountain ridge measurement is almost unaffected by local pollution. It also demonstrates that the influence of local pollution events on the Schneefernerhaus measurements is

successfully removed by the station manager. Only a small deviation up to 0.24 ppm can be observed during the day between the CO$_2$ time series of Schneefernerhaus and the mountain ridge in winter, probably due to anthropogenic sources.

## 1 Introduction

Since the beginning of the industrial era, the mole fraction of greenhouse gases, such as CO$_2$ and CH$_4$, has increased strongly (Etheridge et al., 1996, 1998; Dlugokencky and Tans, 2022; Dlugokencky, 2022). The atmospheric mole fractions of CO$_2$ and

CH$_4$ in preindustrial times are determined by measuring, for example, air enclosed in ice cores (Etheridge et al., 1996, 1998), and since the second half of the 20th century (e.g., Keeling et al., 1976), direct atmospheric measurements of CO$_2$ and CH$_4$ have been possible. Historically, most measurement stations are located on coasts (e.g., Mace Head), island mountains (e.g., Mauna Loa, Izaña) or continental mountains (e.g., Jungfraujoch or Zugspitze), as the measurements done at these stations are typically less affected by regional and local influences than at urban stations. Thus, long-term records from remote stations

play an important role in improving understanding of the global carbon cycle and the impact of greenhouse gases on global warming.

Trace gas measurements of CO$_2$ have been performed at Mount Zugspitze since 1981 at different locations (Reiter, 1986; Yuan et al., 2019). First, at a pedestrian tunnel (ZPT), approximately 250 m below the summit, then at the terrace of the summit (ZUG) and since 2002 at the Environmental Research Station Schneefernerhaus (ZSF). The scientific program at Schnee-





fernerhaus is operated by several German research institutes with the aim of monitoring the physical and chemical properties of the atmosphere and analysing various processes that influence the weather and climate (UFS, 2020). At Schneefernerhaus the measurements of CO, $CO_2$ and $CH_4$ mole fractions are performed by the German Environmental Agency (UBA - Umweltbundesamt). Zugspitze greenhouse gas measurement program is part of the GAW program and joined the atmospheric network ICOS (Integrated Carbon Observation System) in 2021.

Due to the high elevation of mountain stations, the measured ambient air is less affected by regional and local influences compared to urban stations. In recent years, several studies have analysed the $CO_2$ and $CH_4$ measurements performed at Zugspitze (e.g., Yuan et al., 2018, 2019; Giemsa et al., 2019), also in combination with $\delta^{13}$C-$CO_2$ ratios (e.g., Ghasemifard et al., 2019a, b). However, recent studies of greenhouse gas measurements with a temporal resolution of seconds to minutes have shown, that local influences of anthropogenic activities can be seen in several mountain station time series, such as Pic du

Midi, where a small sewage treatment facility near the air intake of the analyser causes local $CH_4$ peaks (El Yazidi et al., 2018), or Jungfraujoch, where the $CO_2$ measurement shows an influence of local anthropogenic activities, potentially by visitors and tourism (Affolter et al., 2021).

Since 2012, new measurement techniques with a temporal resolution of nearly 1 Hz have been used at Schneefernerhaus to measure atmospheric CO, $CO_2$ and $CH_4$ mole fractions. This enabled the detection of local pollution events in the measured

records, which could not be seen to this extent in previous measurements with lower time resolution. In particular, in winter, snow groomers and gasoline snow blowers lead to strong CO peaks, which must be flagged manually to prevent an influence on the records. The German Meteorological Service (DWD - Deutscher Wetterdienst) has also found that its measurements of radon ($^{222}$Rn) activity at Schneefernerhaus are partially contaminated by local geogenic sources of radon. Therefore, since 2014 they have used an ambient air inlet at the mountain ridge (ZGR, 2825 m a.s.l.) approximately 156 m uphill Schneefernerhaus for

their measurements of $^{222}$Rn (Frank et al., 2017). As the local sources influencing the measurements of CO, $CO_2$ and $CH_4$ at Schneefernerhaus are presumably near the station, a change in the location of the air intake line could also reduce the influence on the CO, $CO_2$ and $CH_4$ time series. A new inlet line, installed from Schneefernerhaus to the mountain ridge in 2018, allowed for the measurement of CO, $CO_2$ and $CH_4$ mole fractions in ambient air collected at the mountain ridge since October 2018.

In our study, a two-year long comparison measurement between ambient air of Schneefernerhaus and of the mountain ridge

is analysed. The main focus of this study is to characterise the local pollution events and to compare the influence of these events on the Schneefernerhaus and mountain ridge measurements.

## 2 Methods

### 2.1 Site description

Mount Zugspitze is Germany's highest mountain and its summit is 2962 m above sea level. It is located in southern Germany

in the northern Alps at the border with Austria (Fig. 1). The surrounding area consists largely of bare land, forests and pastures. Urban areas occur less frequently. Around 11 km northeast of Mount Zugspitze lies the city of Garmisch-Partenkirchen





(∼27 000 inhabitants). Innsbruck (∼131 000 inhabitants, 35 km southeast) and Munich (∼1 509 000 inhabitants, 90 km northeast) are the largest cities within 100 km.

Atmospheric greenhouse gas measurements are carried out by the German Environment Agency (UBA) at the Environmental
Research Station Schneefernerhaus (47° 25' 00" N, 10° 58' 46" E), which is approximately 300 m below the summit of Mount Zugspitze (Fig. 1, right panel). The air inlet for the measurements at Schneefernerhaus (ZSF, 2669 m a.s.l.) is installed at the research terrace on the fifth floor (Fig. 1, right panel). The analysis of recent high temporal resolution trace gas measurements has shown that the measured records at Schneefernerhaus are occasionally influenced by local pollution.

Possible sources can be found at Schneefernerhaus itself, at the rack railway tunnel from Schneefernerhaus to the valley, at
the summit of Zugspitze and at the glacier plateau Zugspitzplatt, which is a ski resort in winter. Zugspitzplatt is located approximately 100 m below Schneefernerhaus. Each year more than 500 000 tourists (Bayerische Zugspitzbahn Bergbahn AG) visited the summit of Zugspitze and Zugspitzplatt, which can be reached from the valley by cable cars and rack railways. Furthermore, working activities at Schneefernerhaus, such as the usage of gasoline snow blowers, can influence the measurement. To keep local impacts low, electric snow blowers are generally used. However, after heavy snowfall, gasoline-powered equipment is
needed. On weekdays, 12 people normally work at Schneefernerhaus during regular working hours.

In 2018, the German Meteorological Service (DWD) replaced an old and broken inlet line from Schneefernerhaus to the mountain ridge (ZGR, 2825 m a.s.l.) with a 290 m long new inlet line made of stainless steel (Fig. 1). Since October 2018, it has been possible to simultaneously measure the mole fraction of CO, $CO_2$ and $CH_4$ in the ambient air of Schneefernerhaus and of the mountain ridge above Schneefernerhaus.

## 2.2  Experimental setup

Since October 2018, the mole fractions of CO, $CO_2$ and $CH_4$ in ambient air at Zugspitze have been measured with three analysers, which are installed in the Research Station Schneefernerhaus and are connected to one of two inlet lines. One leads to the research terrace of Schneefernerhaus (ZSF, 2669 m a.s.l.) and the other to the mountain ridge (ZGR, 2825 m a.s.l.). In addition to the ambient air of Schneefernerhaus or the mountain ridge, the analysers simultaneously measure the
same calibration and target gases for quality control. The measuring routine is controlled by the same multiposition rotary valve (model: EMT2SF6MWE, Valco Vici, Switzerland), over which all three instruments receive the measuring samples. The experimental setup is shown in Fig. A1.

The $CO_2$ and $CH_4$ mole fractions in ambient air of Schneefernerhaus are measured with a cavity ring-down spectroscopy (CRDS) G2301 analyser (Picarro, Inc., Santa Clara, CA) and the CO mole fractions with an off-axis integrated cavity output
spectroscopy (OA-ICOS) EP30 analyser (LGR, Los Gatos Research). Ambient air is pumped with a flow rate of 500 L min$^{-1}$ from the research terrace on the fifth floor through the glass inlet at a height of 2.5 m above the terrace. To avoid freezing, the top of the glass inlet is heated. Part of the flow is then passed through a ∼350 ml cold trap (Gaßner Glastechnik GmbH, Germany) to dry the ambient air and, thus, reduce the influence of water vapour on the measurement. The fluid bath of the cold trap is filled with silicon oil (model: M90.055/03, Huber Kältemaschinenbau AG, Germany) and is cooled to −80°C with
a cryogenic cooler (model: TC100E, Huber Kältemaschinenbau AG, Germany). The dried air is then measured by the CRDS





G2301 and the OA-ICOS EP30 analysers, with the flow rate controlled to 0.2 and $0.4 \, \mathrm{L \, min^{-1}}$, respectively. Therefore, the residence time of air in the setup from the inlet at the research terrace to the analysers is only 35 s.

The CO, $CO_2$ and $CH_4$ mole fractions in ambient air of the mountain ridge are measured with a CRDS G2401 analyser (Picarro, Inc., Santa Clara, CA). The air is pumped through a 290 m stainless steel tube (1.2 cm o.d.) from the ridge to Schneefern-
erhaus with a flow rate of $16 \, \mathrm{L \, min^{-1}}$. A part of the air flow is then dried using the same drying system as the one used for the Schneefernerhaus measurements. The residence time of ambient air from the mountain ridge to the analyser is 6 min 40 s. Therefore, to account for the different residence times between the Schneefernerhaus and mountain ridge measurements, the CRDS G2401 data averaged over one minute are shifted by $-6$ minutes.

Ambient air of the mountain ridge was not dried from the beginning (October 2018), as the drying system was installed
on 5 February 2019. The use of the drying system was necessary since the water correction of the CRDS G2401 analyser for CO was insufficient. Therefore, the CO mole fraction measured at the mountain ridge until 5 February 2019 was corrected by an offset of 5.2 ppb. An effect of two pumps installed within the flow path (one for each sampling line) on CO measurements was also noted. Thus, both pumps were relocated out of the direct flow path on 16 July 2019 and the CO mole fractions for both analysers are corrected by $-7.5$ ppb until then. Since the CO measurements of ambient air of Schneefernerhaus and of the
mountain ridge were equally affected, the difference between both measurements was not influenced.

During this study, the UBA air quality measuring station on Zugspitze joined the atmospheric network ICOS (Integrated Carbon Observation System). In the course of the ICOS labelling process, the CRDS G2301 and the OA-ICOS EP30 analysers were sent to the ICOS Atmosphere Thematic Centre (ATC) Metrology Laboratory for validation. Therefore, two large data gaps occurred in the measured time series. From 9 May 2019 to 8 August 2019, no CO, $CO_2$ and $CH_4$ mole fractions were
measured at ZGR and between 1 January 2020 and 8 July 2020, no CO mole fractions were measured at ZSF.

## 2.3 Calibration

The mole fractions of CO, $CO_2$ and $CH_4$ in ambient air of ZSF and ZGR were measured with a temporal resolution of 5 to 10 seconds with two CRDS analysers (G2301, G2401) and one OA-ICOS EP30 analyser. These measurements are averaged over one minute. The minute values of the ambient air time series are calibrated using a two-point calibration to take into account
the drift of the analysers and to link the records to the international scale (CO: WMO X2014A, $CO_2$: WMO X2019, $CH_4$: WMO X2004A). Therefore, every three days, all analysers measure the same low and high working standard cylinders for 15 minutes. Four reference gases provided by the ESRL's Global Monitoring Laboratory (GML) of the National Oceanic and Atmospheric Administration (NOAA) are analysed every two months. These WMO reference gases span the CO mole fraction from 124 to 269 ppb, the $CO_2$ mole fraction from 379 to 430 ppm and the $CH_4$ mole fraction from 1835 to 2120 ppb. Unless
otherwise specified, the calibrated average values over one minute were used in the analysis of the time series.

In addition to calibration cylinders, a target cylinder is simultaneously measured every three days by all analysers for quality control. The averaged calibrated target measurements showed no significant mean difference between the G2301/EP30 and G2401 analyses for all three species CO, $CO_2$ and $CH_4$ (Table 1).





## 2.4   Data flagging

The station manager from UBA manually applied different flags to the Schneefernerhaus CO, $CO_2$ and $CH_4$ time series to document invalid values, such as artefacts and outliers, due to technical problems or work at the setup, as well as local pollution events. Additional data from other trace gas measurements, meteorological data, and station logbooks were used for this purpose. These data from Schneefernerhaus were called "ZSF QC" in this study. However, since this study also aims to assess the impact of local pollution on the Schneefernerhaus time series, a dataset that excludes only instrumental and technical

problems, but not local pollution events, was also used. This dataset was called "ZFS with local pollution". Since the mountain ridge is more elevated than Schneefernerhaus, no strong pollution events in the measured time series were assumed. Thus, only technical artefacts and outliers were flagged in the time series from the mountain ridge, called "ZGR".

## 3    Results and Discussion

### 3.1   CO, $CO_2$ and $CH_4$ mole fractions measured at Schneefernerhaus and the mountain ridge

Since October 2018, CO, $CO_2$, and $CH_4$ mole fractions have been measured in ambient air of Schneefernerhaus and the mountain ridge. Figure 2 shows the time series of 1 min averaged values at both sites. The measurements from the mountain ridge (ZGR) are displayed in red, the ZSF QC time series are shown in black, and the one "with local pollution" in blue. In contrast to the mountain ridge measurement, strong pollution events can be observed at Schneefernerhaus (ZSF time series "with local pollution"). These are mainly visible in the CO record, but also in $CO_2$ and partly in $CH_4$. High CO pollution

events of more than 400 ppb can be noticed, especially during the snow season. These spikes are caused by snow clearing in front of the station with gasoline snow blowers or in preparation of the nearby ski area with snow groomers. In January 2019, heavy snowfall and an avalanche that passed over the station led to intensive use of gasoline snow blowers at the station. During this time, extremely high CO mole fractions of up to 28 000 ppb and corresponding $CO_2$ and $CH_4$ spikes were measured at Schneefernerhaus. In addition, $CO_2$ peaks can be caused by scientists working on the measurement terrace near the inlet. These

local pollution events at Schneefernerhaus were identified and flagged manually by the station manager to reduce and avoid their influence on the measurement.

   The ZSF QC time series had similar CO, $CO_2$ and $CH_4$ mole fractions than measurements from the mountain ridge and both time series followed the same seasonal variations. The CO mole fractions ranged from 48 ppb to 342 ppb. The $CO_2$ mole fractions showed a seasonal cycle with lower values in summer varying between 390 ppm and 440 ppm, and the mole

fractions of $CH_4$ ranged from 1872 ppb to 2100 ppb. As expected, the measurements of ambient air of the mountain ridge and Schneefernerhaus showed similar large scale patterns, but at the mountain ridge, they are much less influenced by local pollution.





## 3.2 Local pollution events of CO and $CO_2$ at Schneefernerhaus

In the CO and $CO_2$ ZSF time series "with local pollution" that averaged over 1 min, large and narrow peaks can occasionally be
detected. Approximately 2 % of CO or $CO_2$ data measured at Schneefernerhaus were strongly contaminated by local pollution
and are therefore flagged manually by the station manager. This corresponds to 1 000 high CO events (12 900 minutely values)
and 2 100 high $CO_2$ events (25 900 minutely values).

Figure 3 shows the number of local polluted and manually flagged CO and $CO_2$ data points against the time of day (a) and
month (b). Due to their anthropogenic origin, strong diurnal and seasonal variations in the occurrence of pollution events are
observed. The polluted and manually marked CO and $CO_2$ data points occurred at 92 % and 95 %, respectively, during the day
between 6 and 18 UTC. Furthermore, 94 % and 92 % of the polluted and manually flagged CO and $CO_2$ data occurred during
the snow season at high altitude (October to May) and after snowfall events when snowblower and snow groomer are used
more often.

In 2020, the COVID-19 pandemic led to temporary restrictions on tourism at Zugspitze and a reduction in work at Schneefern-
erhaus and the ski resort. The resulting decrease in local emissions can be observed in the CO and $CO_2$ time series (Fig. 2), as
well as in $NO_2$, as a reduction in local pollution events.

## 3.3 Local pollution events of $CH_4$ at Schneefernerhaus

The $CH_4$ mole fraction measured at Schneefernerhaus is also occasionally influenced by local pollution events. The high $CH_4$
peaks in the ZSF time series "with local pollution" coincide with high CO or $CO_2$ events and are manually excluded by the
station manager. Furthermore, inverse $CH_4$ spikes of several ppb occurred in the Schneefernerhaus time series, which were not
detected at the mountain ridge. Typically, these inverse spikes do not occur as single events, but during periods with multiple
inverse spikes. These periods can last between some hours to multiple days and start and end abruptly (see Fig. 4). Several tests
performed during such events with inverse $CH_4$ spikes show that these spikes are neither an artefact nor caused by the analyser,
the measurement setup, or the inlet. Thus, $CH_4$ depleted air must be transported occasionally to the measuring terrace.
From the Schneefernerhaus station, a rack railway tunnel leads downhill to the valley. An opening to this tunnel is located
45 m from the measurement terrace to direct the tunnel air with high radon concentration to the outside and to reduce the radon
concentration inside the Schneefernerhaus station. For 15 days in November 2020, the $CH_4$ mole fraction was measured inside
the tunnel near the opening. The measured mole fraction of $CH_4$ strongly depends on the direction of the tunnel air flow. As
the tunnel air flows downhill, fresh air enters the tunnel through the opening, and the $CH_4$ mole fraction has only slightly
180 lower values than those measured on the terrace. However, extremely low $CH_4$ mole fractions between 500 and 1000 ppb are
measured when the tunnel air flows uphill. In this case, local wind pattern can transport the $CH_4$-depleted tunnel air into the
ambient air inlet line at Schneefernerhaus, resulting in inverse $CH_4$ spikes. Since the tunnel air flows downhill more frequently
in summer than in winter, the influence of the tunnel air, and thus of the inverse $CH_4$ spikes, seems to be larger in winter.





### 3.4 Comparison of local pollution events at Schneefernerhaus and the mountain ridge

Although strong local pollution events are noticeable in the CO and $CO_2$ mole fractions measured at Schneefernerhaus, such strong events did not occur in the mountain ridge time series. Only small peaks are sometimes noticeable in the mountain ridge record, when the wind blows from southeast, from the station Schneefernerhaus to the mountain ridge (see Appendix B). However, these pollution events have much smaller amplitudes than those measured simultaneously at Schneefernerhaus. On approximately 83 % of the days when local pollution events were visible in the CO or $CO_2$ measurements at Schneefernerhaus, we did not find corresponding peaks at the mountain ridge. Even extremely high CO events of more than 1 000 ppb at Schneefernerhaus are usually not detectable at the mountain ridge.

Figure 5 shows frequency distributions of deviations between the Schneefernerhaus and mountain ridge time series averaged to minute values for CO (Fig. 5 a) and $CO_2$ (Fig. 5 b) mole fractions. On the left, the Schneefernerhaus time series "with local pollution" are used for the calculation and on the right the ZSF QC dataset, where local pollution events were manually excluded by the station manager. The daytime data (red) show a high number of large positive differences when the ZSF time series "with local pollution" are used. These strong positive deviations disappear on the right panels when the ZSF QC data are included in the calculation. This indicates that the large positive differences are caused by local pollution at Schneefernerhaus. Furthermore, it shows that strong local pollution events in the Schneefernerhaus time series are successfully excluded by the station manager.

### 3.5 Differences between Schneefernerhaus and the mountain ridge measurements

The above analyses of the Schneefernerhaus and the mountain ridge measurements have shown that the mountain ridge time series are less influenced by local pollution, and strong local pollution events at Schneefernerhaus are successfully flagged by the station manager. To verify whether there are further influences from local sources in the Schneefernerhaus time series, which do not manifest themselves as spikes, the hourly mean values of the measurements from Schneefernerhaus were compared with those from the mountain ridge. These correspond to the QC datasets reported to national and international databases.

Figure 6 shows the hourly averaged mole fraction of CO, $CO_2$ and $CH_4$ measured at Schneefernerhaus (black) and at the mountain ridge (red), as well as the difference between both locations. The hourly data for Schneefernerhaus used in this comparison are calculated out of the ZSF QC time series in which local pollution events are manually excluded. The average difference between Schneefernerhaus and mountain ridge data is $0.4 \pm 2.4$ ppb for CO, $0.1 \pm 0.4$ ppm for $CO_2$ and $-0.4 \pm 3.4$ ppb for $CH_4$ and meets the compatibility goal determined by the WMO (2020). The large standard deviations in CO, $CO_2$ and $CH_4$ differences cannot be explained by instrumental noise. A comparison of the CO, $CO_2$ and $CH_4$ mole fractions measured with OA-ICOS EP30 or CRDS G2401 and CRDS G2301 using exactly the same air shows a standard deviation of hourly averages of 0.6 ppb CO, 0.02 ppm $CO_2$ and 0.2 ppb $CH_4$. Thus, the large standard deviations of the hourly differences between the ambient air of Schneefernerhaus and the mountain ridge are probably caused by the different measuring locations. As local wind conditions can strongly vary between Schneefernerhaus and the mountain ridge (see Appendix B and Fig. B1), air masses often occur with a time delay between the two locations.





To analyse the distribution of the differences between the ZSF QC and the mountain ridge measurement in more detail, the frequency of the one minute differences between both locations are shown on the right side in Fig. 6. In addition to the whole dataset (black), the distribution is shown for nighttime (blue, 18 to 6 UTC) and daytime (red) mole fractions. The differences
in the CO mole fractions show a small shift of 0.3 ppb between nighttime and daytime and a symmetric distribution. However, the differences are still within the uncertainties of the analysers.

The $CO_2$ mole fractions show larger differences during the day, with an average of 0.13 ppm. This is approximately 0.07 ppm larger than that during the night and exceeds the WMO compatibility goal of 0.1 ppm (WMO, 2020). Furthermore, the distribution is asymmetric to positive differences, which even increase during the day. This indicates elevated $CO_2$ mole fractions
at Schneefernerhaus at daytime values. One explanation for this may be that the influence of local sources is not completely excluded by manual flagging, but the different sites may also receive slightly different signals from regional pollution due to the different altitudes and complex topography. The influence of local sources on the $CO_2$ time series at remote mountain station is noticed also at other stations. At the mountain station Jungfraujoch, the average difference between the usual location and another, less polluted one, is 0.01 ppm overnight and even 0.49 ppm during the day (Affolter et al., 2021).
In contrast to the $CO_2$ mole fractions, $CH_4$ time series from Schneefernerhaus and the mountain ridge are comparable within the WMO compatibility goal of 2 ppm (WMO, 2020) overnight and during the day. However, the distribution is slightly asymmetric towards negative values. This distribution is probably related to inverse $CH_4$ spikes visible at Schneefernerhaus, which are caused by $CH_4$-depleted air from the rack railway tunnel (Sect. 3.3).

## 3.6  Annual and diurnal cycles

The annual and diurnal cycles of the mole fraction of CO, $CO_2$ and $CH_4$ measured in ambient air at Schneefernerhaus and the mountain ridge are analysed and compared in Fig. 7. To describe the Schneefernerhaus data the ZSF QC dataset (black) and the ZSF time series "with local pollution" (blue) are used again. In the calculations, only data for which simultaneous measurements at both locations are available are taken into account. The diurnal cycles for the high alpine winter (October to March) and from April to September are determined by calculating the diurnal variations per month and then averaging them.
The general pattern of the annual and diurnal cycles of CO, $CO_2$ and $CH_4$ measured at Schneefernerhaus and the mountain ridge are similar to those of other mountain stations in the Northern Hemisphere. The cycles are mainly determined by the planetary boundary layer height. Thus, depending on the season and the time of day, the air masses of the free troposphere or the planetary boundary layer are measured. Since air from the planetary boundary layer is more polluted in CO and $CH_4$, higher values are measured in summer and especially during the day. Since $CO_2$ uptake by the biosphere reduces the $CO_2$ mole
fraction in the boundary layer, the $CO_2$ mole fraction measured at Zugspitze is lower in summer and during the day.

In the following, the difference between the Schneefernerhaus and mountain ridge measurements is discussed in more detail. The monthly averaged CO and $CH_4$ mole fractions measured at Schneefernerhaus (ZSF QC) and the mountain ridge (ZGR), as well as the diurnal cycles for both locations, do not show differences below the recommended compatibility of 2 ppb. For $CH_4$, a small annual cycle in the differences can be noticed, with slightly positive values in summer and negative values in
winter. The especially low $CH_4$ mole fractions at Schneefernerhaus between November 2018 and January 2019 compared to





the mountain ridge are caused by particularly frequent and strong inverse $CH_4$ spikes in the Schneefernerhaus time series (see Sect. 3.3).

The monthly $CO_2$ mole fractions measured at Schneefernerhaus are comparable to those at the mountain ridge in summer. However, stronger deviations are observed in winter. In addition, the diurnal $CO_2$ cycle at Schneefernerhaus shows up to
0.24 ppm higher values in the afternoon than at the mountain ridge, although the strong local pollution events in the Schneefernerhaus time series are already excluded. Otherwise, the difference would be up to 0.35 ppm in winter. The difference between the diurnal cycles at Schneefernerhaus and at the mountain ridge in the afternoon decreases in summer compared to winter, but it does not disappear completely. Thus, the ZSF QC time series still show influences from local or regional sources in winter and during the day, which are not visible at the mountain ridge due to the higher elevation, complex wind pattern (see
Appendix B and Fig. B1), and the topography of the Zugspitze.

Other mountain stations also show local influences, which depend on the season and time of day. At Jungfraujoch, the diurnal cycle of $CO_2$ between measurements at the usual location and a less polluted site deviate up to 0.4 ppm in winter and up to 1.3 ppm in summer (Affolter et al., 2021). Thus, the $CO_2$ measurement is more strongly influenced in summer, which is probably due to the larger number of tourists. At Schneefernerhaus, on the other hand, the strongest influence was observed in
winter.

### 3.7 Weekend dependency in the Schneefernerhaus time series

In this study, a small influence, especially on the CO and $CO_2$ time series, was noticed due to anthropogenic activities throughout the day. In addition, approximately 92 % of data points which were excluded manually due to local pollution occur on weekdays and not on weekends. In a previous study by Yuan et al. (2019), which analysed the $CO_2$ time series at Schneeferner-
haus between 2002 and 2016, a weekly cycle in the $CO_2$ measurement with lower $CO_2$ mole fractions on weekends and higher $CO_2$ mole fractions on weekdays is described. Additionally, they found stronger diurnal cycles on weekdays with higher values, especially in the morning, for the combined time period 2002 to 2016. They explained these findings by local anthropogenic working activities at Schneefernerhaus.

When analysing the Schneefernerhaus and mountain ridge measurements between October 2018 and October 2020 for the
weekly cycle and stronger diurnal cycles, no weekend dependencies were detected for either the $CO_2$ or the CO and $CH_4$ data. Therefore, we investigated the $CO_2$ time record for Schneefernerhaus between 2002 and 2021, similar to Yuan et al. (2019). A statistical analysis was performed using the resampling technique described in Daniel et al. (2012) to determine if a weekly cycle showed a significant pattern. No significant weekend effect (mean weekend value minus mean weekday value) occurred until 2007, and after 2014. However, significantly higher values were evident during the week in the data between
2008 and 2014 (Fig. C1). Furthermore, the average diurnal $CO_2$ cycles on weekends were comparable for the three time periods 2002–2007, 2008–2014 and 2015–2021, which is not the case for weekdays. A strong difference of up to 1.5 ppm was noticed between the diurnal $CO_2$ cycles on weekends and weekdays in the time periods 2002–2007 and 2008–2014 (Fig. C2). Such a difference is not noticeable in the $CO_2$ data from 2015 to 2021.





A micro leakage in the inlet system could explain the noticed weekend effect and the higher $CO_2$ values during the day on weekdays, as the strongest effect would be noticeable during the working hours when the $CO_2$ mole fraction in the laboratory is higher. Between 2014 and 2015, the entire measurement setup, including the intake system, was thoroughly checked for leakages, and old connections were renewed. In the process, the high temporal resolution $CO_2$ measurements of the CRDS analyser were also used. Presumably, the micro leakage was fixed at that time, as no weekend dependency has been detectable since then.

## 4   Conclusions

Ambient air measured at Schneefernerhaus with high temporal resolution showed local pollution events caused by human activities, especially in CO and $CO_2$. These peaks occur mainly in the winter season and during the day. Therefore, approximately 2 % of the measured data were flagged manually by the station manager.

To prevent the influence of this local pollution, additional measurements were taken at a more elevated location for comparison. This study showed that the atmospheric mole fractions of CO, $CO_2$ and $CH_4$ can be measured at the mountain ridge using a 290 m long intake line. The time series determined in this way does not show a strong influence of local pollution events, unlike the measurements at Schneefernerhaus.

The comparison of measurements performed at Schneefernerhaus with background time series from the mountain ridge characterises the local influence on the CO, $CO_2$ and $CH_4$ time series. It is confirmed that strong local pollution events at Schneefernerhaus can be successfully removed from the Schneefernerhaus time series by the station manager. No significant difference in the annual cycle of CO and $CH_4$ for the Schneefernerhaus time series with manually flagged pollution events and the mountain ridge could be noticed, and the diurnal cycles were similar at both locations, too. The annual and diurnal cycles in $CO_2$ measured at Schneefernerhaus and the mountain ridge also show the same pattern. In summer the difference between monthly values and diurnal cycles is negligible. In winter, however, higher $CO_2$ mole fractions occur at Schneefernerhaus especially during the day, which indicates a small impact of local or regional pollution that is not yet excluded.

With the exception of slightly elevated $CO_2$ mole fractions during the day, the CO, $CO_2$, and $CH_4$ measurements at the Schneefernerhaus station are also consistent with background measured at the mountain ridge. However, as it is difficult to maintain the 290 m intake line, especially in the winter months, and thus to ensure continuous series of measurements on the mountain ridge, it is important to continue measurements at both sites in the future.

*Data availability.* The high resolution (1 min) CO, $CO_2$ and $CH_4$ records of Zugspitze-Schneefernerhaus (ZSF) and the mountain ridge (ZGR) used in this study are available on request from the data owner (Cedric.Couret@uba.de). Hourly averaged CO, $CO_2$ and $CH_4$ records of Zugspitze-Schneefernerhaus (ZSF) are available from the World Data Centre for Greenhouse Gases (WDCGG) at https://gaw.kishou.go.jp/.





**Appendix A:  Experimental setup at Schneefernerhaus**

**Appendix B:  Wind pattern**

The wind direction and wind velocity are measured by DWD at Schneefernerhaus, at the mountain ridge and at Zugspitze summit. Figure B1 shows the wind roses computed from the available wind direction and the wind velocity at the three locations between October 2018 and October 2020. The colours indicate different wind speed ranges. The wind direction measured at Schneefernerhaus and at the mountain ridge is most likely influenced by the characteristic locations and thus by the topography of the mountain. The mountain ridge above Schneefernerhaus runs largely from west to east, so Schneefernerhaus is on the

southern slope of the mountain. Therefore, north and south winds are more frequently measured on the mountain ridge and the main wind directions at Schneefernerhaus are east and west. At the Zugspitze summit, the inflow of air masses is less restricted, with more frequent winds from the northwest, southwest and southeast. Furthermore, the average wind velocity at the summit is $6.0\,\mathrm{m\,s^{-1}}$ and therefore larger than that at the mountain ridge ($4.0\,\mathrm{m\,s^{-1}}$) or Schneefernerhaus ($3.2\,\mathrm{m\,s^{-1}}$).

**Appendix C:  Weekend dependency in the Schneefernerhaus time series between 2002 and 2021**

To determine if the $CO_2$ mole fraction is generally higher on weekdays than on weekends, the data were detrended by subtracting a moving 31-day mean to remove the annual cycle. Since we are interested in the average variations within a week, weeks with missing days were excluded. A statistical analysis was performed using the resampling technique described in Daniel et al. (2012) to identify if a weekly cycle shows a significant pattern. For each year, we tested whether the weekend effect magnitude (mean weekend value minus mean weekday value) was significant. Until 2007, no significant ($p > 0.1$) weekend effect was no-

ticeable. Between 2008 and 2014 every year except 2013 showed a significant ($p < 0.1$) weekend effect. No weekend effect has been visible since 2015. Therefore, we separate the $CO_2$ time series into three parts: 2002–2007, 2008–2014 and 2015–2021. Figure C1 shows the weekly cycles for all three time intervals. Even visually, the difference between the weekly cycles for 2008–2014 and the other two time periods is obvious. As expected, the weekly cycles for the time intervals 2002–2007 and 2015–2021 show no significant ($p < 0.05$) weekend effect, as does the weekly cycle for the time period 2008–2014.

Furthermore, we calculated the mean diurnal cycles for weekends and weekdays for each year. Before averaging, each single diurnal cycle is subtracted by the moving 31-day average and the average nighttime (18 to 6 UTC) $CO_2$ mole fraction for each week to remove seasonal variations. For the years 2002 to 2014, a similar weekend dependency can be noticed as reported by Yuan et al. (2019), with much stronger diurnal cycles on weekdays than on weekends. Since 2015, the difference in the diurnal cycles between weekdays and weekends disappeared. Figure C2 shows the diurnal cycles for weekdays and weekends for the

winter (October to March) and summer (April to September). The weekday (Monday to Friday) diurnal cycles are shown in red and weekend cycles (Saturday and Sunday) in black. In addition, the difference between averaged weekday and weekend values are shown. For the first two time periods (2002–2007 and 2008–2014), stronger $CO_2$ peaks (5 to 15 UTC) were noticed during weekdays, which did not occur on weekends. The difference between weekdays and weekends was lower than $0.2\,\mathrm{ppm}$ during the night and reached values between 1 and $1.5\,\mathrm{ppm}$ during the day. In the time period 2015–2021 the diurnal cycles of



weekends and weekdays showed no strong differences, with values lower than 0.2 ppm for the whole day. Comparing the three

periods, it was further noticed that the diurnal cycles of weekends have the same amplitude and shape, especially in summer.

Yuan et al. (2019) explained the weekend dependencies by local anthropogenic working activities at Schneefernerhaus. This

seems plausible, since the main increase in $CO_2$ occurred during working hours on weekdays. The tourist activities around

Schneefernerhaus are, on the other hand, normally stronger on weekends than on weekdays. However, the absence of the

weekly cycle and of the difference in the diurnal cycles of weekday and weekend data since 2015 implicate a relevant change

in the working activities at Schneefernerhaus that did not occur. It is more likely that a micro leakage in the inlet system

caused higher $CO_2$ values during the week until it was fixed between 2014 and 2015. The micro leakage affected the measured

$CO_2$ mole fractions only during working hours, when the $CO_2$ mole fraction inside the laboratory was much higher due to

the presence of humans. The analysis has shown that nowadays there is no strong impact of local pollution on weekends or

weekdays.

*Author contributions.*    AH and MS designed the study together with CC. AH evaluated the data and wrote the paper with the help of MS, CC
and BH. CR was responsible for the GHG measurements at Scheefernerhaus and at the mountain ridge.

*Competing interests.*    The authors declare that they have no conflict of interest.

*Acknowledgements.*    The authors would like to thank the German Meteorological Service (DWD - Deutscher Wetterdienst) for providing the
intake line to the mountain ridge. We also thank Gabriele Frank (DWD) and Josef Salvamoser (IGU - Institut für angewandte Isotopen-, Gas-
und Umweltuntersuchungen) for making it possible to use the intake line to the mountain ridge for our comparison measurements.

The data analysis of this study was funded by Projects 95297 and 167847 with the German Environment Agency (UBA). We wish to thank
Frank Meinhardt and Ludwig Ries for their great support in these cooperation projects between the UBA and Heidelberg University.





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





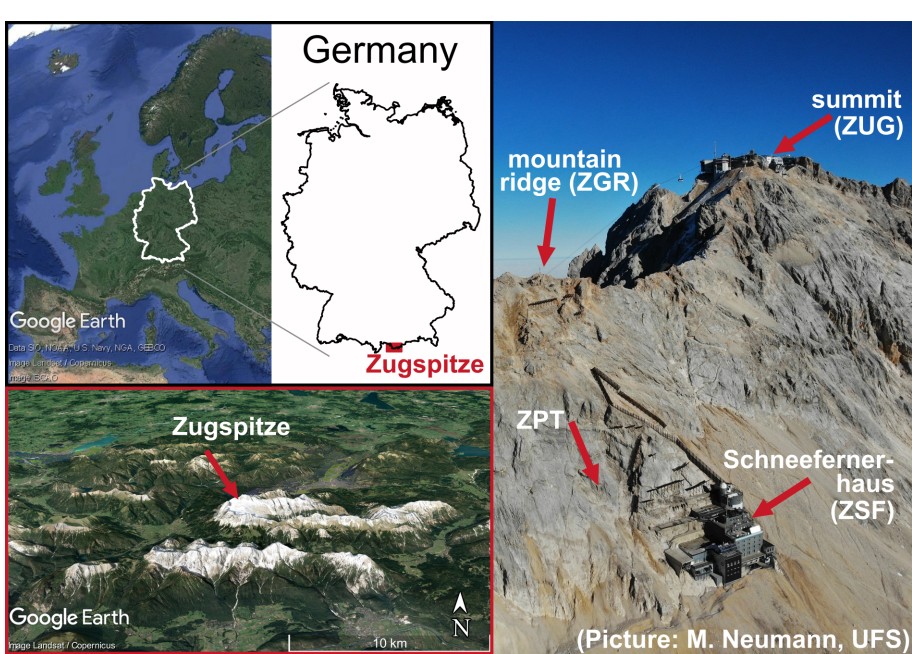

**Figure 1.** Location of the Zugspitze-Schneefernerhaus measurement site and of the two inlet lines at Schneefernerhaus and the mountain ridge (Map data on left side: from © Google Earth).





**Table 1.** Calibrated target gas measurements were performed with the CRDS G2301, G2401 and OA-ICOS EP30 analysers. The average and standard deviation of each analyser and the differences are shown.

| analyser | CO [ppb] | $CO_2$ [ppm] | $CH_4$ [ppb] |
|---|---|---|---|
| CRDS G2301 | - | $404.87 \pm 0.03$ | $1887.10 \pm 0.23$ |
| CRDS G2401 | $124.6 \pm 0.8$ | $404.87 \pm 0.02$ | $1887.07 \pm 0.03$ |
| OA-ICOS EP30 | $124.6 \pm 0.1$ | - | - |
| | | | |
| difference | $0.03 \pm 0.87$ | $-0.001 \pm 0.017$ | $0.03 \pm 0.23$ |

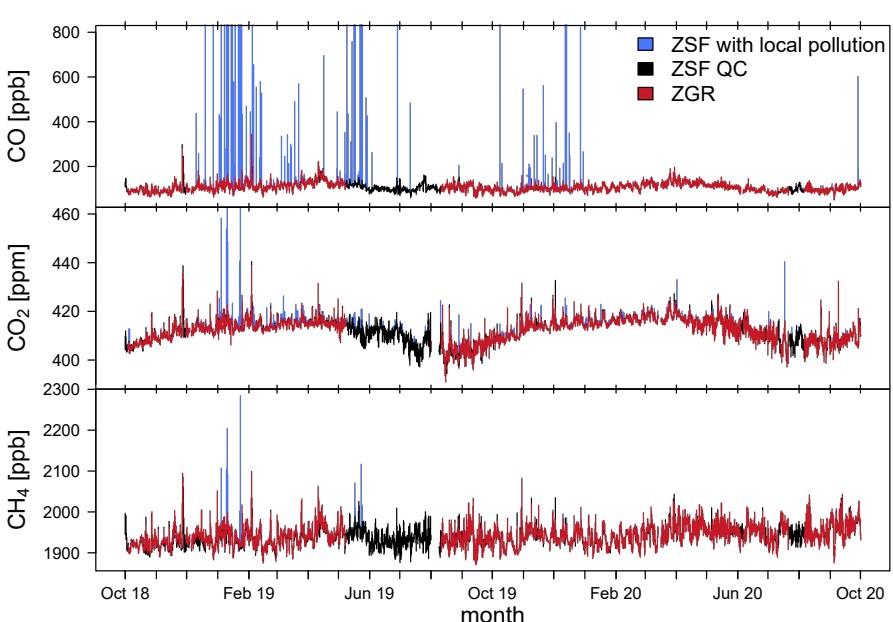

**Figure 2.** CO, $CO_2$ and $CH_4$ mole fractions in ambient air of Schneefernerhaus (ZSF QC: black, ZSF with local pollution: blue) and of the mountain ridge (red). The shown data are averaged over 1 min.





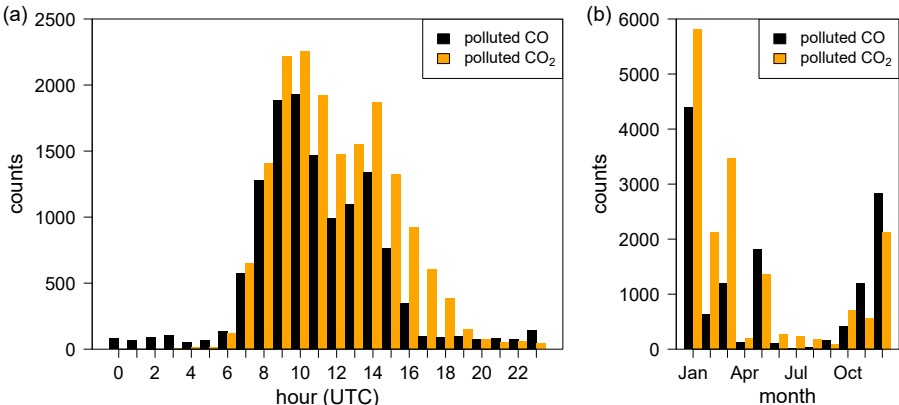

**Figure 3.** Distribution of local polluted and manually flagged CO and $CO_2$ data at Schneefernerhaus as a function of month and time of day. The 1 min averaged dataset was used for this analysis.



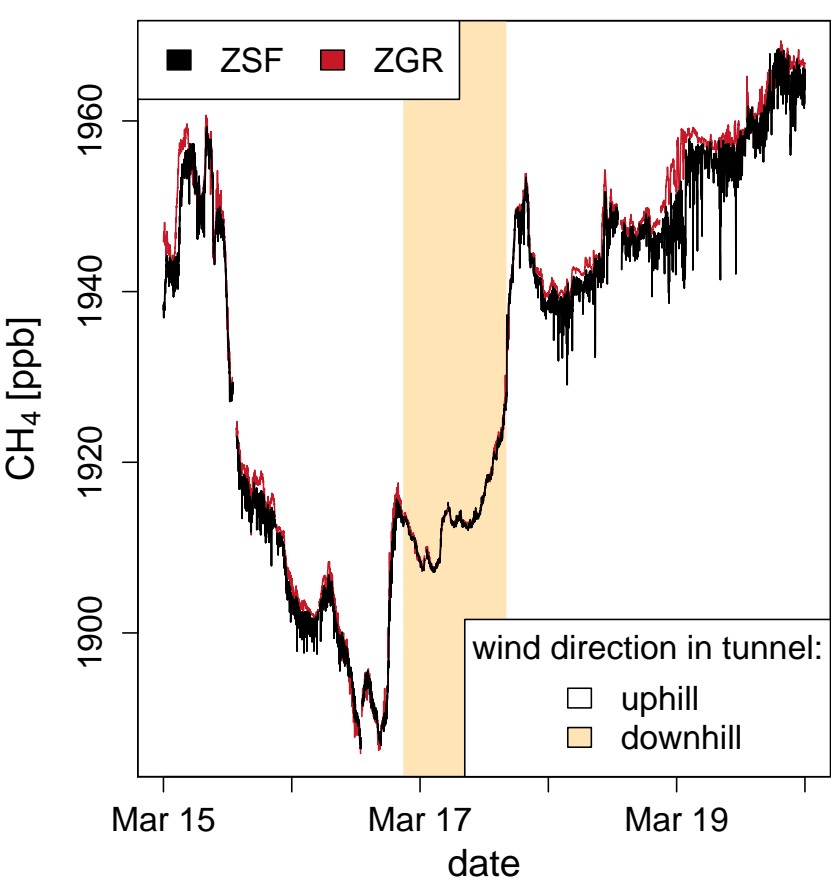

**Figure 4.** Example of inverse $CH_4$ spikes measured at Schneefernerhaus (black) but not at the mountain ridge (red). The beige-coloured time interval corresponds to a downhill wind-direction in the rack railway tunnel.





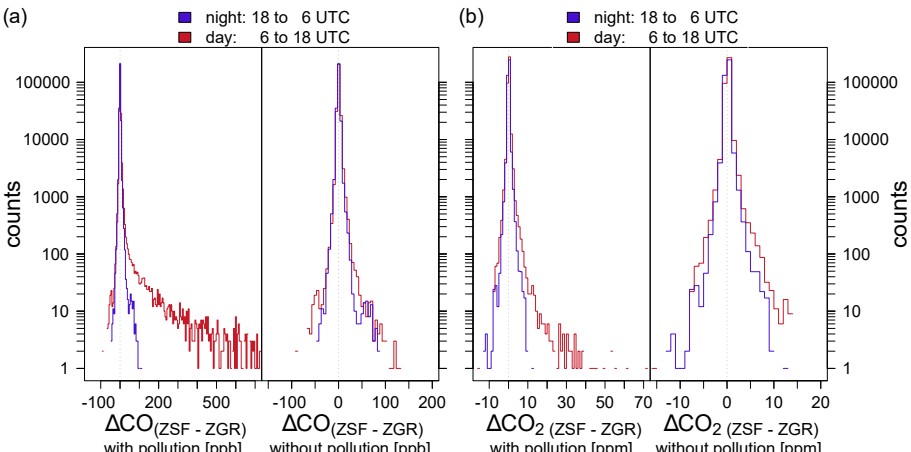

**Figure 5.** The frequency distribution of the difference between the 1 min averaged Schneefernerhaus and the mountain ridge measurements for CO (a) and $CO_2$ (b). The counts are displayed on a logarithmic scale. The blue colour corresponds to nighttime (18 to 6 UTC) and red to daytime (6 to 18 UTC) data.



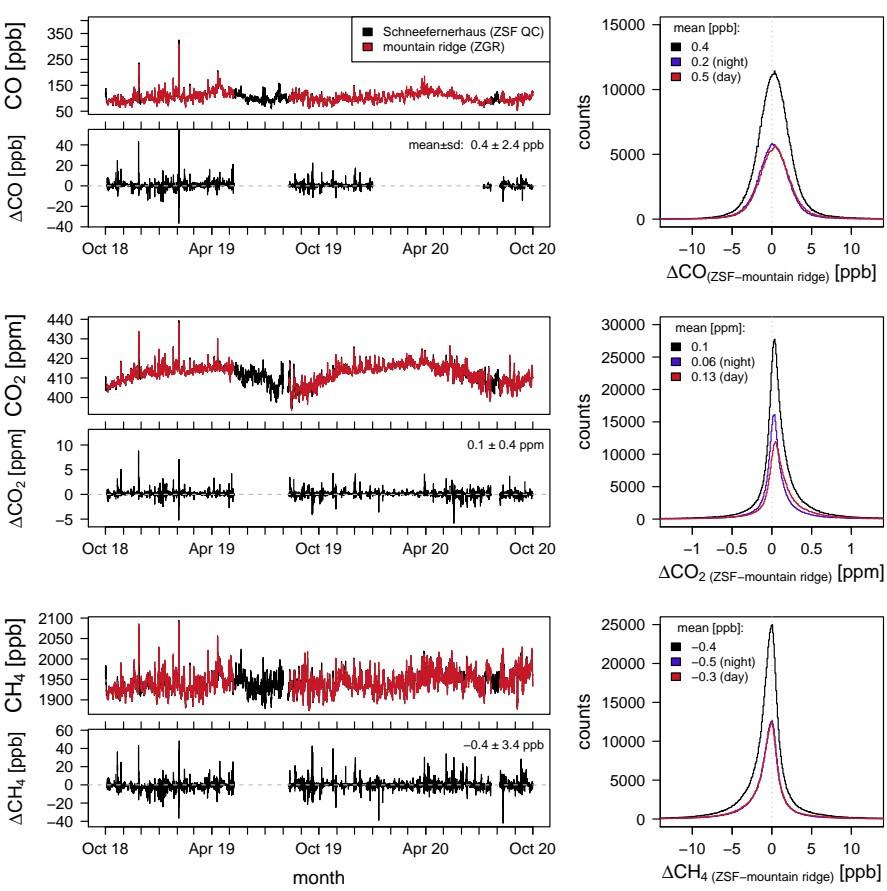

**Figure 6.** Left: Hourly averaged ZSF QC and ZGR mole fractions of CO, $CO_2$ and $CH_4$, as well as the difference between both locations. Right: The frequency of the differences between Schneefernerhaus and the mountain ridge calculated using the minutely averaged data.



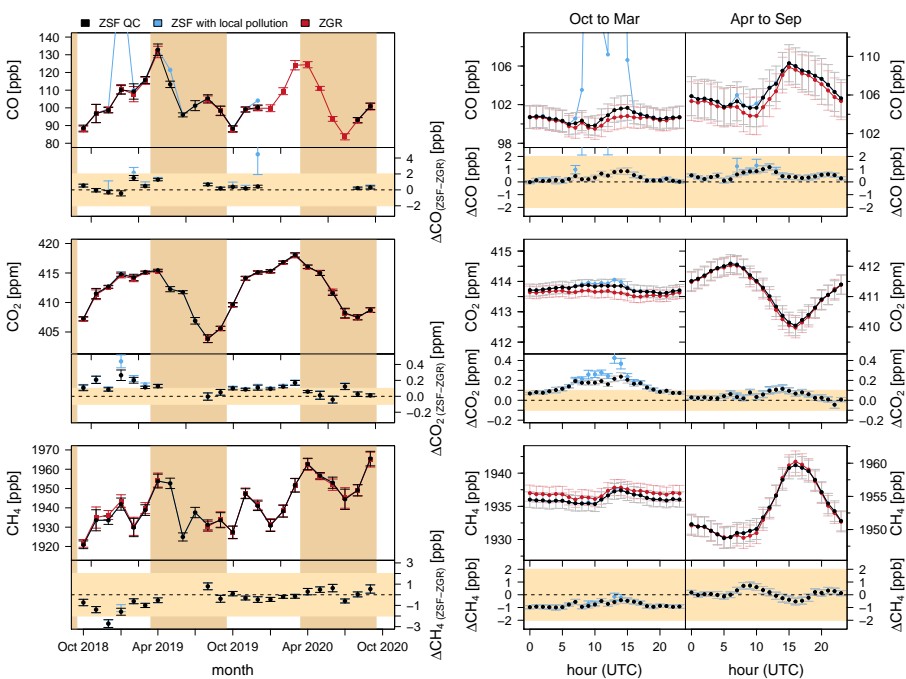

**Figure 7.** Annual and diurnal cycles in the Schneefernerhaus (black/blue) and mountain ridge (red) measurements. In addition, the difference between the Schneefernerhaus and mountain ridge data is shown.


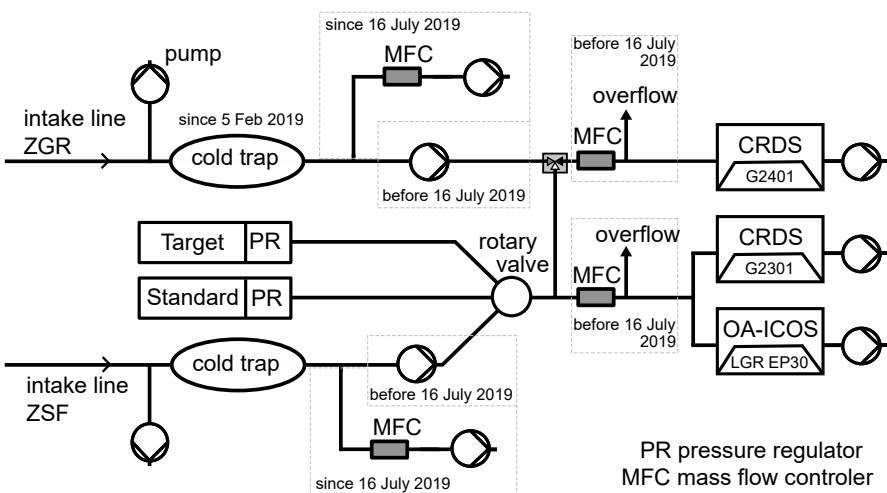

**Figure A1.** Experimental setup to measure CO, CO$_2$ and CH$_4$ in ambient air from Schneefernerhaus and from the mountain ridge.





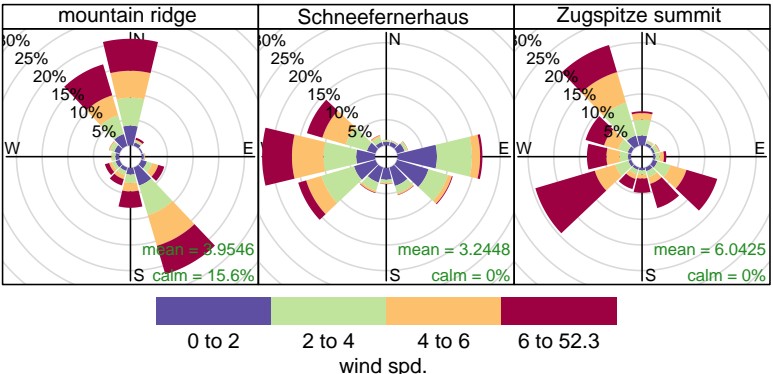

**Figure B1.** Wind rose plots for the mountain ridge (left), Schneeferneerhaus (middle) and Zugspitze summit (right) showing the wind directions and wind velocities between October 2018 and October 2020 provided by DWD. The colours correspond to different wind speed ranges.

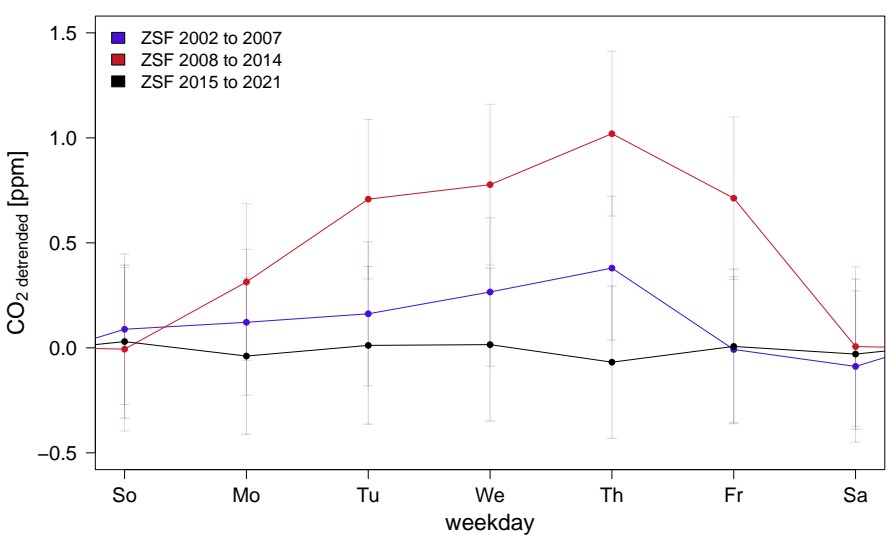

**Figure C1.** Weekly cycles in $CO_2$ measured at Schneefernerhaus during three time periods: : 2002–2007 (blue), 2008–2014 (red), 2015–2021 (black). The $CO_2$ mole fraction is detrended by subtracting the moving 31-day average.

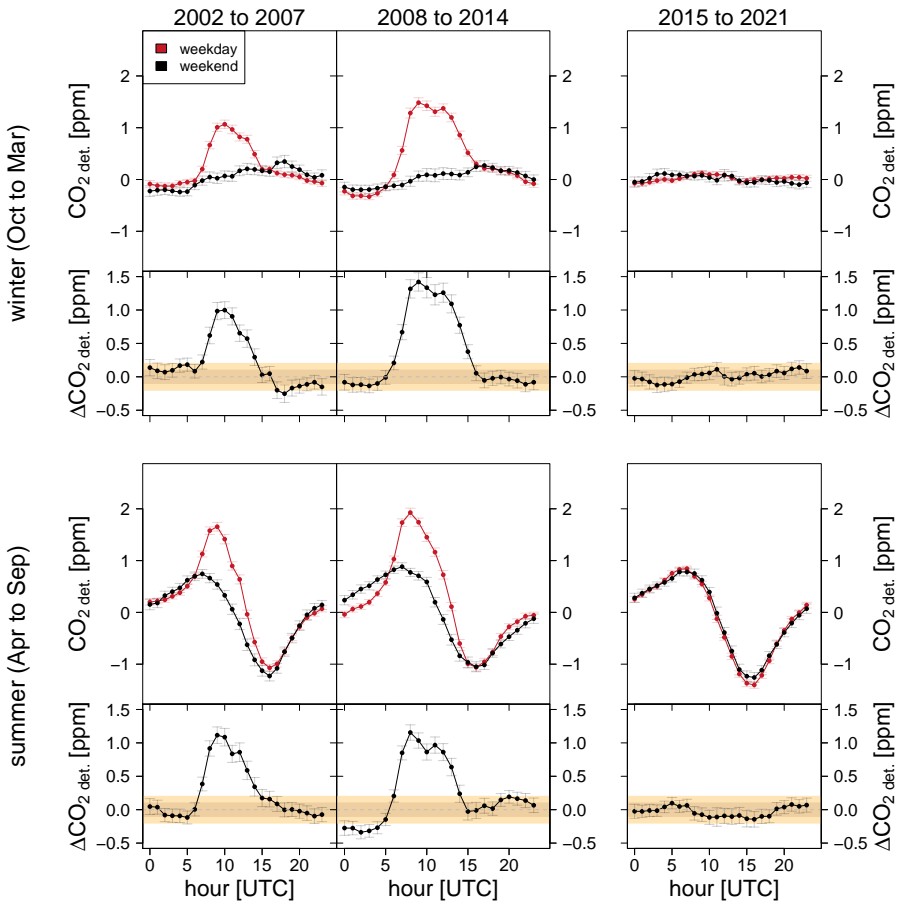

**Figure C2.** Mean diurnal cycles for Schneefernerhaus calculated from the weekend and weekday data between 2002 and 2021 for two seasons. Diurnal cycles calculated using weekday data are coloured red, and those using weekend data are in black. The top panels correspond to the seasons October to March, and the bottom panels correspond to April to September. The cycles in the lower part of the two panels show the difference between weekdays and weekends.