# Peer review of "Comparison of atmospheric CO, CO2 and CH4 measurements at Schneefernerhaus and the mountain ridge at Zugspitze"

_Atmospheric Measurement Techniques, 2022_

## Author Response (AR1)

**Author's Response to Referee Comments on:** "Comparison of atmospheric CO, $CO_2$ and $CH_4$ measurements at Schneefernerhaus and the mountain ridge at Zugspitze

We thank the editor and the reviewers for their detailed and constructive comments and their useful suggestions. We have revised the manuscript accordingly.

**General Comments:**

[Referee #1]     The authors attribute the $CO_2$ spikes to anthropogenic contamination, namely in the abstract. Is it possible to attribute those spikes only to anthropogenic sources? Or the complex topography and/or wind patterns may also play an important role as mentioned (Lines 259-260). Maybe that the authors should stress the difficulty they encountered when disentangling both processes.

[Hoheisel et al.]     Thank you for pointing this out. The $CO_2$ spikes, which are then flagged by the station manager, are most certainly caused by local anthropogenic sources such as snow cats, snow blowers and humans. In addition to these identifiable $CO_2$ spikes, we can observe a difference in $CO_2$ levels between the mountain ridge and Schneefernerhaus in the data (especially in the mean diurnal cycles). This difference occurs especially during the day and in winter. As discussed in chapter 3.5 and 3.6, we cannot clearly explain the underlying cause of this difference. The most likely scenarios are that the known local $CO_2$ sources (in this case anthropogenic) cause the increase and their influence could not be completely removed by filtering the peaks, as the influence is not recognisable as spikes. On the other hand, the fact that the elevation occurs mainly in winter during the day also suggests that $CO_2$-enriched air is transported from the valley to the Schneefernerhaus and that, due to the elevated position of the mountain ridge and the complicated topography, the mixing is not large enough to reach the mountain ridge. In the second consideration, most of the additional $CO_2$ comes from more regional $CO_2$ sources. These could be, for example, anthropogenic emissions from heating.

[Referee #1]     Data flagging: I understand that CO and CH4 spikes can be easily flagged (essentially due to snow clearing); however, how are the CO2 spikes manually flagged? Except for scientists working on the terrace, is there any other CO2 contamination flagged? Is it possible to quantify the influence of scientists working on the terrace?

[Hoheisel et al.]     CO, $CH_4$ and $CO_2$ spikes, are in principle flagged in the same way. For this purpose, as described in chapter 2.4, the station manager uses meteorological data, other trace gases such as NO and $NO_2$ and the station logbook in which, for example, work at Schneefernerhaus and on the measurement terrace is recorded. In addition to $CO_2$ spikes caused by scientists and workers, $CO_2$ contamination also occurs during the use of snow clearing equipment.

[Referee #1]     The paper would benefit in general from a few more appropriate references. As for instance in Section 3.6.

[Hoheisel et al.]     We added a few addition references on different sections of the paper. However, in section 3.6 we only discuss the $CO_2$ data of Schneefernerhaus and the mountain ridge, which we measured in this study. Therefore, we see no need for additional references here.

[Referee #1]     The figure captions could be extended with more detailed information so that we can understand them alone.

[Hoheisel et al.]     We appreciate the recommendation and we have improved and expanded the figure captions so that they can be understood on their own

**Detailed Comments and Questions:**

[Referee #1]    Lines 17-18: Add countries to station's name

[Hoheisel et al.]    We included the countries to the station's name and as suggested by Referee#2 also references for the cited stations. We changed the sentence to:
*"Historically, most measurement stations are located on coasts, such as Mace Head, Ireland (Bousquet et al.,1996), island mountains, such as Mauna Loa, Hawaii (Keeling et al., 1976) or Izana, Spain (Navascues and Rus, 1991; Gomez-Pelaez et al., 2019) or continental mountains, such as Mount Cimone, Italy (Ciattaglia, 1983), Jungfraujoch, Switzerland (Sturm et al., 2005; Schibig et al., 2016) or Schauinsland, Germany (Schmidt et al., 2003)."*

[Referee #1]    Line 37: and tourism

[Hoheisel et al.]    Unfortunately, the intention of this comment is not entirely clear to us. The second part of the sentence of line 37 is: *"…, such as Pic du Midi, where a small sewage treatment facility near the air intake of the analyser causes local $CH_4$ peaks (El Yazidi et al., 2018), or Jungfraujoch, where the $CO_2$ measurement shows an influence of local anthropogenic activities, potentially by visitors and tourism (Affolter et al., 2021)."*

[Referee #1]    Lines 41-42: Is there a local reference paper? You may want to cite Bukowiecki et al., 2021. **DOI** 10.1088/2515-7620/abe987

[Hoheisel et al.]    Thanks to the comment, we realised that we had not expressed ourselves clearly in the manuscript. With the sentence in lines 41-42, we did not intend to make a general statement, but to draw attention to the influence of snow groomers and gasoline snow blowers that we found in the Schneefernerhaus data, which in our case are noticeable as CO peaks. We changed the sentence to: *"In particular, in winter, snow groomers and gasoline snow blowers lead to strong CO peaks in the Schneefernerhaus time series, which must be flagged manually to prevent an influence on the records."*

[Referee #1]    Line 44: Isn't it 126 m higher?

[Hoheisel et al.]    We have checked the difference in altitude between mountain ridge and Schneefernerhaus again. The actual sampling altitude at the mountain ridge is 2825m a.s.l. and the sampling altitude at Schneefernerhaus for CO, $CO_2$ and $CH_4$ is 2669 m a.s.l. Thus, we came up with 156m. Since $^{222}Rn$ at Schneefernerhaus is not measured with the same intake line than CO, $CO_2$ and $CH_4$ and the exact elevation of the intake at the mountain ridge probably changed since 2014, after a new intake line was installed, we have relativised the sentence to:
*„Therefore, since 2014 they have used an ambient air inlet at the mountain ridge approximately 150m uphill Schneefernerhaus for their measurements of $^{222}Rn$ (Frank et al., 2017)."*

[Referee #1]    Line 61: You mention an elevation of 2669m a.s.l. for the air inlet located at the 5th floor. In the abstract, you mention an elevation of 2666m a.s.l. for the station. Is it correct?

[Hoheisel et al.]    We have checked the elevations again. The air inlet for the CO, $CO_2$ and $CH_4$ measurements at Schneefernerhaus is installed at the research terrace on the fifth floor at an elevation of 2669 m above sea level.
The Schneefernerhaus altitude is given as 2650 m above sea level, so we changed the value in the abstract. The former value of 2666m a.s.l. we had written in the abstract is the elevation of the Schneefernerhaus lab, where CO, $CO_2$ and $CH_4$ are measured.

[Referee #1]    Lines 63-64. A reference is missing here. Or you mean that it is shown in this paper?

[Hoheisel et al.]    Yes, we meant that it is shown in our paper. To clarify, we changed *"The analysis"* to *"Our analysis"* in line 63.

| [Referee #1] | Lines 73-74: add ZSF and ZGR. Throughout the text you mostly use "Schneefernerhaus" and only a few times "ZSF" (for instance lines 110, 112, 154). Stick maybe to "Schneefernerhaus". The same with "mountain ridge" and sometimes "ZGR" (lines 110,111). |
|---|---|
| [Hoheisel et al.] | Thank you, we had not noticed that before. We have now tried to stick to the words "Schneefernerhaus" and "mountain ridge". However, in some cases it is necessary to give more precise information, such as "ZSF QC", in order to clarify exactly which dataset is being used. |

| [Referee #1] | Lines 164-166. Can you mention the restriction? Did it lead to the cease of tourism for a while? It could also be interesting to show on Fig. 2 from when to when these restrictions occurred. The covid pandemics could possibly be discussed and used in this study to a greater extent, as it is supposed to have noticeably reduced pollution events. |
|---|---|
| [Hoheisel et al.] | Yes, tourism on the Zugspitze was temporarily suspended and work at Schneefernerhaus and the ski resort was reduced to a minimum. During the COVID-19 pandemic restrictions at the time of our comparison measurements, it was unfortunately not possible for us to measure CO in the ambient air of Schneefernerhaus, as our measuring device was being validated at the ICOS Atmosphere Thematic Centre (ATC) Metrology Laboratory as part of the ICOS labelling process at that time. Therefore, we were not able to carry out an evaluation of the influence of the restriction on our measurements as detailed as we had hoped. Instead of CO, we thus examined $NO_2$ and $CO_2$ and were able to determine a reduced occurrence of spikes in the measured time series. Later restrictions at the Zugspitze, which no longer fall within the time frame of our comparison measurements, also showed a lower number of CO peaks at the Schneefernerhaus due to the restrictions. However, as these measurements are not part of our comparison measurements, we have decided to discuss the effects of the COVID-19 restrictions only briefly. We have revised the relevant section again to avoid inconsistencies and misunderstandings. |

*"In 2020, the COVID-19 pandemic led to temporary restrictions on tourism at Zugspitze and a reduction in work at Schneefernerhaus and the ski resort. Unfortunately, our CO analyser for ambient air from Schneefernerhaus was being validated at the ICOS ATC Metrology Laboratory as part of the ICOS labelling process at that time. Instead, we analysed $NO_2$ measured at Schneefernerhaus. Especially for $NO_2$ but also for $CO_2$ a decrease in the occurrence of $NO_2$ and $CO_2$ peaks was observed in the measured time series at Schneefernerhaus, indicating a reduction of local pollution events."*

| [Referee #1] | Line 170: Is it a CH4 reverse "spike" or more generally a longer excursion? |
|---|---|
| [Hoheisel et al.] | Yes, these are reverse\inverse $CH_4$ spikes as explained in more detail in section 3.3 and are shown in Figure 4. |

| [Referee #1] | Line 205: "to the QC datasets reported to national and international databases". Not clear, precise. |
|---|---|
| [Hoheisel et al.] | Yes, we have excluded this sentence in the text and added another one in section 2.4 for better understanding. |

*"The half-hourly, hourly, daily and monthly averages of CO, $CO_2$ and $CH_4$ mole fractions at Schneefernerhaus reported to national and international databases such as the World Data Centre for Greenhouse Gases (WDCGG) are also based on one-minute averaged data where local pollution events have been flagged as invalid in addition to artefacts and outliers."*

| [Referee #1] | Lines 264-265: Shouldn't it then be explained by other mechanisms compared to Jungfraujoch? |
|---|---|
| [Hoheisel et al.] | We have probably not been entirely clear in this section. In the paragraph above lines 264-265 we discussed our observations and their causes at Schneefernerhaus. Then, as a comparison, we refer to the observations as well as their causes at Jungfraujoch. We have now made minor changes in the text to make this clearer. The last sentence should then highlight the differences between the influences at the Zugspitze and at the Jungfraujoch. |

*"Other mountain stations also show local influences, which depend on the season and time of day. At Jungfraujoch, the diurnal cycle of $CO_2$ between measurements at the usual location and a less polluted site deviate up to 0.4ppm in winter and up to 1.3ppm in summer (Affolter et al., 2021). Thus, the $CO_2$ measurement at Jungfraujoch is more strongly influenced in summer, which is probably due to the larger number of tourists. At Schneefernerhaus, on the other hand, the strongest influence was observed in winter."*

| | |
|---|---|
| [Referee #1] | Lines 267-268: Are there other possible explanations? |
| [Hoheisel et al.] | We have changed the paragraph slightly in order to clarify our statement. The CO and CO2 spikes found in the time series at Schneefernerhaus are most certainly caused by local anthropogenic sources such as snow cats, snow blowers and humans. |

*"In this study, a more frequent occurrence of CO and $CO_2$ spikes during the day was found, which can be attributed to anthropogenic activities. In addition to the diurnal dependence, these local pollution events also have a week dependence. Thus, 92% of the data points that were manually excluded due to local pollution occur on weekdays and not on weekends."*

**General Comments:**

| | |
|---|---|
| [Referee #2] | As a general comment, I would recommend the authors to provide specific recommendations to the user of the ZSF dataset. As an instance, can potential user (still) use the daytime data from 2002 to 2014 in front of the possible impact of a potential leak in the sampling system? Can you provide recommendation for the fitness of this dataset for specific purposes (e.g. trend calculation, use in inversion modeling, model validation, etc etc)? Similar recommendation should be mirrored in distribution databases like the WDCGG.

 As concerning the ZSF dataset in the WDCGG, it is not completely clear to me if the data available by the WDCGG are the ZSF QC or the ZSF with local pollution (with suitable flags). A clarification to potential users can be really useful. |
| [Hoheisel et al.] | Thank you for pointing this out. We agree that this is important. However, in our study we focus on the comparison between the ambient air measurements from Schneefernerhaus and from the mountain ridge. In this comparison, no difference was found between weekends and weekdays in terms of diurnal cycles or mean values for the individual days of the week. As these findings do not agree with the results of Yuan et al. (2019), we analysed the $CO_2$ time series between 2002 and 2021 with regard to these two effects. We can show that the differences between weekend and weekday no longer occur since 2015 and thus also during our comparison measurement. A more detailed analysis of the long-term $CO_2$ time series between 2002 and 2021 with regard to trend, annual variation and the use of the data for inversion experiments or model validation was not carried out in this comparison study of the two sampling locations. However, in the next update of the data in the WDCGG it is planned to point out the differences between weekdays and weekends and to recommend interpreting the daily values with caution. Regarding the last comment, we have added the following sentence to section 2.4 Data flagging and correction for clarification: |

*"The half-hourly, hourly, daily and monthly averages of CO, $CO_2$ and $CH_4$ mole fractions at Schneefernerhaus reported to national and international databases such as the World Data Centre for Greenhouse Gases (WDCGG) are also based on one-minute averaged data where local pollution events have been flagged as invalid in addition to artefacts and outliers."*

**Detailed Comments and Questions:**

ABSTRACT:

| | |
|---|---|
| [Referee #2] | Line 2: please provide the geographical location of Zugspitze |
| [Hoheisel et al.] | We now have included the geographical location: |

*"The CO, CO$_2$ and CH$_4$ mole fractions have been measured since 2002 at the Environmental Research Station Schneefernerhaus, which is located approximately 300m below the summit of Mount Zugspitze in the German Alps."*

| | |
|---|---|
| [Referee #2] | Line 11: please, be more specific: what anthropogenic sources? |
| [Hoheisel et al.] | We understand the referee's comment. However, in contrast to the CO, CO$_2$ and CH$_4$ peaks where the sources are identifiable, in the case of the CO$_2$ difference in the mean diurnal cycles between Schneefernerhaus and mountain ridge in winter, the exact cause cannot be determined with absolute certainty. The different theories are explained in more detail in the sections 3.5 and 3.6 and involve local or regional anthropogenic sources. In the abstract, we have therefore decided to refer only generally to anthropogenic sources to keep it short. |

**INTRODUCTION**

| | |
|---|---|
| [Referee #2] | Line 17-18: please provide references for the cited stations. Among the continental mountain site also Mt. Cimone (active since 1979 can be cited). |
| [Hoheisel et al.] | We included the references for the cited stations and Mount Cimone. In addition, we included the countries to the station's name as suggested by Referee#1. We changed the sentence to:
*"Historically, most measurement stations are located on coasts, such as Mace Head, Ireland (Bousquet et al.,1996), island mountains, such as Mauna Loa, Hawaii (Keeling et al., 1976) or Izana, Spain (Navascues and Rus, 1991; Gomez-Pelaez et al., 2019) or continental mountains, such as Mount Cimone, Italy (Ciattaglia, 1983), Jungfraujoch, Switzerland (Sturm et al., 2005; Schibig et al., 2016) or Schauinsland, Germany (Schmidt et al., 2003)."* |

| | |
|---|---|
| [Referee #2] | Line 28: please specify GAW acronym |
| [Hoheisel et al.] | Yes, we changed it to "Global Atmosphere Watch (GAW) program". |

**EXPERIMENTAL SET-UP**

| | |
|---|---|
| [Referee #2] | Please, can you provide more description about the management of the calibration and target gas injections? Looking to the setup reported in figure A1, it seems that they are simultaneously sampled by all the three instruments. Is it correct? Did you intercompare the instrument when simultaneously sampling ambient air from the ZSF inlet? |
| [Hoheisel et al.] | Yes, it is correct, that the calibration and target gases are simultaneously sampled by all three analysers. The working standards and the target are measured every three days for 15 minutes and the WMO reference gases of NOAA every two months for 30 minutes. A more detailed description of the usage of calibration and target gases is given in section 2.3 Calibration and quality control. To improve the reading flow, we have now noted this in the text in section 2.2. Experimental setup:
*"In addition to the ambient air of Schneefernerhaus or the mountain ridge, the analysers simultaneously measure the same calibration and target gases for quality control (see Sect. 2.3 for further details)."*
We also compared the CO, CO$_2$ und CH$_4$ mole fractions in ambient air from the Schneefernerhaus inlet measured simultaneously with all analysers for 15 days in July 2020. The average difference between the hourly averaged CO, CO$_2$ and CH$_4$ mole fractions measured with the CRDS G2301&OA-ICOS EP30 and CRDS G2401 are -0.04±0.58 ppb (mean±sd) in CO, 0.04±0.02 ppm in CO$_2$ and 0.3±0.2 ppb in CH$_4$. We included this comparison in the manuscript in section 2.3 Calibration and quality control, too. |

| | |
|---|---|
| [Referee #2] | How the air inlet at ZGR is designed? Is it equipped with heater, rainguard, filters? |
| [Hoheisel et al.] | The air inlet at ZGR is not heated, has no filter but is protected from rain. We included: *"The air inlet at the mountain ridge is protected from rain, but not heated."* |

[Referee #2]   Line 94: specify the inner diameter, I think it is more important than the o.d.
[Hoheisel et al.]   We agree that it would be helpful to specify the inner diameter. As soon as we get the value, we will change this if still possible.

[Referee #2]   Line 95: "a part of the air flow is dried using the same drying system as the ..", Why only "a part"? I cannot see an overflow before the cold trap (figure A1).
[Hoheisel et al.]   Figure A1 shows the setup only schematically, but it shows that the inlet line is purged by a pump. A small part of the air is taken from the main inlet line via a T-piece and purged via the cold trap.

[Referee #2]   Line 98: if the residence time is 6':40", why was the data shifted by 6 minutes and not 7 minutes? Do you monitor the flow stability over the time?
[Hoheisel et al.]   The residence time from the intake at Schneefernerhaus to the analyser is 35s and from the mountain ridge 6min and 40s. Therefore, we shifted the minutely averaged mountain ridge data by 6 minutes, to compare them with the Schneefernerhaus data. The air flow through the long intake line up to the mountain ridge is not monitored.

[Referee #2]   Line 101 – 104. All this section need a more robust explanation and discussion. How did you assess the offset of 5.2 ppb (which is rather large, indeed)? Did you perform an instrument characterization for the impact of the water vapor influence (e.g. https://doi.org/10.5194/amt-5-2555-2012)? The impact of the not efficient water vapor correction must change as a function of the water vapor levels. Why did you not apply a correction function based on the actual water vapor value? How the overall measurement uncertainty was affected by using this fix offset? The same points are valid for the offsets related with the pump within the flow path. Which was the reason of the artifacts by the 2 pumps? Did you note any impact on CO2 and CH4? All these points should be clarified in the text.
[Hoheisel et al.]   We appreciate the recommendation and include a more detailed description of the corrections in section 2.3 Calibration, correction and quality control.
If the pump is installed in the direct flow path, the membrane used in the pump causes the CO mole fraction to increase.

CALIBRATION

[Referee #2]   In general, how often the NOAA calibration gases have been reassigned? How stable were the target gas measurements? Are you able to provide a quantification of the uncertainty based on the target gas results? Please, provide a better description of the calibration strategy (how many cycles, injection duration …). Providing this information will represent a valuable source of info for interested readers.
[Hoheisel et al.]   We appreciate the recommendation. The same four NOAA calibration gases were used for the duration of the two-year comparison measurement. They are measured every two months for 30 minutes each (one cycle). We have also added more details in the paragraph on target gas measurement:
*"In addition to calibration cylinders, a target cylinder is simultaneously measured every three days by all analysers for quality control. The calibrated target measurements of CO, $CO_2$ and $CH_4$ were stable over time and there is no significant mean difference between the G2301/EP30 and G2401 analyses for all three species. Table 1 shows the average and standard deviation of the CO, $CO_2$ and $CH_4$ mole fractions for the two-year comparison for each analyser and their difference."*

RESULTS AND DISCUSSION

[Referee #2]    Line 166: I would expect that very local pollution events like those related with the use of snowblower were traced better by NO peaks (rather than NO2). Can you comment on this?

[Hoheisel et al.]    At Schneefernerhaus, the UBA measures NO and $NO_2$ in the ambient air. In both time series, the use of snow blowers or snow groomers is clearly visible in the form of measurement spikes and an increasing trend during the day.

[Referee #2]    Line 172: which kind of specific tests have been performed? Are you able to completely rule out the possibility of icing on the sampling inlet affecting the sampling efficiency? Please, explain more.

[Hoheisel et al.]    We are glad to provide you with further details on the tests carried out. To exclude the possibility that these inverse spikes are caused by the analyser or the setup, two tests were carried out. First, during an event in which inverse spikes are measured at Schneefernerhaus, the two analysers G2301 and G2401 are exchanged. Thus, analyser G2301 first measures the ambient air of the Schneefernerhaus and later that of the mountain ridge. Analyser G2401 first measures the ambient air of the mountain ridge and later that of the Schneefernerhaus. In all measurements at the Schneefernerhaus, inverse spikes occur for both analysers, while in the measurements at the mountain ridge, no spikes are detected for either analyser. Therefore, analyser G2301 is not responsible for the occurrence of inverse $CH_4$ spikes in the time series at Schneefernerhaus. In a second test, CRDS analyser G2301 measures the ambient air at Schneefernerhaus using the normal setup described in Section 2.2. However, the CRDS analyser G2401 uses a completely different setup with a stainless steel intake line that starts at the same point as the glass intake line for the normal setup. The $CH_4$ mole fractions measured with both analysers but different setups show strong inverse $CH_4$ spikes. Therefore, the experimental setup is not responsible for the inverse $CH_4$ spikes, too.
We also added further details in the manuscript:
*"Two tests were performed during such events with inverse $CH_4$ spikes. First, we have exchanged the analysers and second, we have used an independent intake line that receives ambient air from the same location on the terrace at Schneefernerhaus. The inverse $CH_4$ spikes were measured in air from both sampling lines to the Schneefernerhaus terrace and with both analysers when measuring ambient air from the Schneefernerhaus. However, they were not detected when measuring ambient air from the mountain ridge. Therefore, these spikes are neither an artefact nor caused by the analyser, the measurement setup or the inlet."*
Regarding the possible impact of icing at the sampling inlet, we can exclude this as a cause, since on the one hand the intake at the Schneefernerhaus is heated and on the other hand the air flow through the analyser did not show any irregularities during the occurrence of the spikes.

[Referee #2]    Line 177: this is an interesting experiment. How did you measure the direction of the flow inside the tunnel? I would suggest to change Figure 4 with an example from the experiment in Nov 2020 showing the CH4 near the tunnel entrance with simultaneous measurements at ZSF. Did the CO2 vary during the inverse CH4 peaks?

[Hoheisel et al.]    The flow direction inside the tunnel was measured with a 1D anemometer (of the company Thies) which was installed a few meter within the tunnel. We now have added the instrument type: "*For 15 days in November 2020, the $CH_4$ mole fraction was measured with a CRDS G1301 analyser (Picarro, Inc., Santa Clara, CA) inside the tunnel near the opening. The measured mole fraction of $CH_4$ strongly depends on the direction of the tunnel air flow, which was measured in the tunnel with a 1D anemometer.*"
We appreciate the recommendation regarding Figure 4. In order to continue to give a good idea of the inverse $CH_4$ spikes, we have not replaced Figure 4, but included a figure which shows the measurements in the tunnel in more detail.
So far we have not detected any peaks in the $CO_2$ time series that correlate with the inverse $CH_4$ spikes. This was to be expected, as the $CO_2$ measurements in the

tunnel have shown that the $CO_2$ mole fraction in the tunnel entrance does not change much between periods when air is blowing uphill or downhill.

| | |
|---|---|
| [Referee #2] | Line 205: from a user perspective, it is interesting to know the relationship between the local pollution event flagged in this work with the flags reported in the WDCGG datasets (e.g. Valid (background): 1, Valid (background): U, Invalid: N, Valid (background): O, Invalid: K, Valid (background): R, Invalid: H, Invalid: 3, see https://gaw.kishou.go.jp/search/file/0071-6031-1001-01-01-9999). |
| [Hoheisel et al.] | We can understand that this could be interesting from the user's point of view. Until joining ICOS and thus at the time of the comparison measurements, the one-minute averages were flagged. As described in section 2.4, artefacts, outliers and local pollution events are flagged as invalid and are not taken into account when averaging the one-minute averages to hourly averages. In the WDCGG, the hourly values are then reported as valid or invalid, as appropriate. |
| [Referee #2] | Line 213: how long the intercomparison was? |
| [Hoheisel et al.] | The comparison was done for 15 days. We included the duration in the manuscript: "A comparison of the CO, $CO_2$ and $CH_4$ mole fractions measured for 15 days with OA-ICOS EP30 or CRDS G2401 and CRDS G2301 using exactly the same air shows a standard deviation of hourly averages of 0.6ppb CO, 0.02ppm $CO_2$ and 0.2ppb $CH_4$" |
| [Referee #2] | Line 231: please correct "ppb". |
| [Hoheisel et al.] | Thanks, we changed it. |
| [Referee #2] | Figure 7: please explain what the error bars represent. |
| [Hoheisel et al.] | The error bars are the standard errors of the averages. |
| [Referee #2] | Line 242: I suppose that also thermal valley and slope winds play a role. This should be cited. |
| [Hoheisel et al.] | Yes, also thermal valley and slope winds play a role and we included it: *"The diurnal cycles are mainly determined by the planetary boundary layer height and convective upslope winds (Yuan et al., 2019).* |
| [Referee #2] | Line 244: this is actually not true. For CH4 and CO the seasonal peaks are occurring in spring (March-May). During the summer a relative minimum is evident likely due to lower combustion emissions (for CO) and enhanced OH removal (for CO and CH4). |
| [Hoheisel et al.] | Yes, in our effort to briefly explain several processes, we wrote the paragraph inaccurately and misleadingly. We have now changed the paragraph to: *"The general pattern of the annual and diurnal cycles of CO, $CO_2$ and $CH_4$ measured at Schneefernerhaus and the mountain ridge are similar to those of other mountain stations in the Northern Hemisphere (Thoning et al., 1989; De Wekker et al., 2009). The diurnal cycles are mainly determined by the planetary boundary layer height and convective upslope winds (Yuan et al., 2019). Depending on the season and the time of day, the air masses of the free troposphere or the planetary boundary layer are measured. Since air from the planetary boundary layer is typically more polluted in CO and $CH_4$, higher values are measured during the day especially in summer. Since $CO_2$ uptake by the biosphere reduces the $CO_2$ mole fraction in the boundary layer, the $CO_2$ mole fraction measured at Zugspitze is lower during the day in summer."* |
| [Referee #2] | Line 258: I think that the higher deviation during winter daytime (when vertical transport from the PBL is minimized) can be a point towards local influences. |
| [Hoheisel et al.] | That's right, we also assume that the difference between the $CO_2$ measurement at Schneefernerhaus and at the mountain ridge is due to the reduced vertical transport in winter. However, we have not been able to clarify whether the higher $CO_2$ mole fraction at Schneefernerhaus is due to local sources in the surroundings of Schneefernerhaus, or whether it is due to more regional sources such as heating, whereby the $CO_2$-enriched air masses come up the valley but not all the way to the mountain ridge. |

[Referee #2]     Line 279 – 284. I'm confused. Figure C2 reported a significant deviation between weekday and weekend also in 2002-2007. Why is not this evident in Figure C1?

[Hoheisel et al.]     Yes, also in the period 2002-2007 differences between the diurnal cycles at the weekend and during the week can be seen in Figure C2. However, these are not quite as large on average as between 2008 and 2014. In Figure C1, one can also notice a small tendency towards higher values during the week for 2002-2007 compared to the period 2015-2021. However, the changes are so small that the significance test for weekly dependence does not detect a significant weekend effect for the period 2002-2007. Furthermore, we would like to note that in order to determine the mean diurnal cycles, detrended values were used, i.e. in our case the mean night value was subtracted for each day. In Figure C2, therefore, mean weekday differences are not completely included.

[Referee #2]     The presence of a leakage in the sampling system in the period 2002-2014 can potentially have impact on the trend calculation and (considering that the impact is exceeding 1 ppm during daytime hours) on utilization of this dataset for inversion experiments (even if modelers are usually taking night-time data from mountain sites) or model validation. I would like to see recommendations to users about the use of this earlier ZSF dataset. These recommendations should be also provided in the national/international database using these data.

[Hoheisel et al.]     We agree that this is an important point. We have answered these comments of the referee in the context of the first general comment of the referee at the beginning.

CONCLUSIONS:

[Referee #2]     Line 298: "baseline" should be used instead of "background"
[Hoheisel et al.]     Yes, we changed it to "baseline".

APPENDIX C:

[Referee #2]     Line 326: how the 31-day moving window was defined? Did you perform sensitivity tests changing the length of the time window?
[Hoheisel et al.]     For each day, a 31-day moving average was calculated by averaging the daily means in the period of 15 days before and 15 days after that day.

---

## Author Response (AR2)

**Author's Response to Referee Comments on:** "Comparison of atmospheric CO, CO$_2$ and CH$_4$ measurements at Schneefernerhaus and the mountain ridge at Zugspitze

We thank the editor and the reviewers for the second review of this manuscript and their constructive revisions and their useful suggestions. We have revised the manuscript accordingly.

| | |
|---|---|
| [Referee #2] | This is my second review of this manuscript by Hoheisel et al. I recognise that the authors addressed in a good way all the points that I reported in the review. I can only suggest a few further revisions before the publication of this interesting paper (please refer to the Author's tracked changes document)
1) page 5, line 146: 5 ppm -> 5 ppb |
| [Hoheisel et al.] | Thanks for pointing it out, we have of course corrected it. |

| | |
|---|---|
| [Referee #2] | 2) Page 5, line 142 - 148: Thanks to the authors for providing details about the test performed to evaluate the impact of WV to CO. However, the authors should be aware that the influence of WV to CO measurements by CRDS is erratic and can significantly vary on time (see e.g. https://doi.org/10.5194/amt-14-89-2021, section 4.5). I suggest that the authors comment on the manuscript about the limitation of their approach (i.e. "one-shot" test, no information about the dependency of the CO impact as a function of WV values). |
| [Hoheisel et al.] | We have included a sentence about the limitations of our approach.
*Since there is no information about the exact dependence of the CO mole fraction as a function of water vapour, the applied CO correction is based on the previously described comparison measurement, which was performed only once.* |

| | |
|---|---|
| [Referee #2] | 3) Page 12, line 379 -382: I suggest to clearly state (as the authors nicely did in the Author's Response) that also in the period 2002-2007 differences between the diurnal cycles for weekend and during the week can be detected even if so small that that the significance test for weekly dependence did not detect a significant weekend effect. |
| [Hoheisel et al.] | Thank you for this comment. We have included some further clarifications.
*Even visually, the difference between the weekly cycles for 2008–2014 and the other two time periods is obvious. As expected, the weekly cycle for the time interval 2015–2021 shows no significant (p<0.05) weekend effect, as does the weekly cycle for the time period 2008–2014. A slight tendency towards higher values during the week can also be observed for the 2002-2007 time interval compared to the 2015-2021 period. However, the variation is so small that the significance test (p<0.05) shows no significant weekend effect for the period 2002-2007.* |